# TRPC and TRPV Channels’ Role in Vascular Remodeling and Disease

**DOI:** 10.3390/ijms21176125

**Published:** 2020-08-25

**Authors:** Marta Martín-Bórnez, Isabel Galeano-Otero, Raquel del Toro, Tarik Smani

**Affiliations:** 1Department of Medical Physiology and Biophysics, University of Seville, 41009 Seville, Spain; mmartin55@us.es (M.M.-B.); igaleano@us.es (I.G.-O.); rdeltoro-ibis@us.es (R.d.T.); 2Group of Cardiovascular Pathophysiology, Institute of Biomedicine of Seville, University Hospital of Virgen del Rocío/University of Seville/CSIC, 41013 Seville, Spain; 3Biomedical Research Networking Centers of Cardiovascular Diseases (CIBERCV), 28029 Madrid, Spain

**Keywords:** TRPC, TRPV, vascular tone, vascular disease, vascular remodeling

## Abstract

Transient receptor potentials (TRPs) are non-selective cation channels that are widely expressed in vascular beds. They contribute to the Ca^2+^ influx evoked by a wide spectrum of chemical and physical stimuli, both in endothelial and vascular smooth muscle cells. Within the superfamily of TRP channels, different isoforms of TRPC (canonical) and TRPV (vanilloid) have emerged as important regulators of vascular tone and blood flow pressure. Additionally, several lines of evidence derived from animal models, and even from human subjects, highlighted the role of TRPC and TRPV in vascular remodeling and disease. Dysregulation in the function and/or expression of TRPC and TRPV isoforms likely regulates vascular smooth muscle cells switching from a contractile to a synthetic phenotype. This process contributes to the development and progression of vascular disorders, such as systemic and pulmonary arterial hypertension, atherosclerosis and restenosis. In this review, we provide an overview of the current knowledge on the implication of TRPC and TRPV in the physiological and pathological processes of some frequent vascular diseases.

## 1. Introduction

Blood vessels are composed essentially of two interacting cell types: endothelial cells (ECs) from the tunica intima lining of the vessel wall and vascular smooth muscle cells (VSMCs) from tunica media of the vascular tube. Blood vessels are a complex network, and they differ according to the tissue to which they belong, having diverse cell expressions, structures and functions [1,2,3]. Blood vessels’ heterogeneity is further enhanced by pathophysiological stimuli [4], showing different characteristics and behaviors to certain diseases such as atherosclerosis [5], hypertension [6], restenosis [7] or thrombosis [8]. Nevertheless, within the variability, there are common features that define blood vessels. Inside vessels, the blood flow and pressure are coordinated by multiple mechanisms, which can be classified into extrinsic ones, including hormonal and neuronal regulation, and intrinsic ones through myogenic and metabolic regulation [9]. Local intrinsic factors are able to activate different intracellular pathways to stimulate the required physiological response. Most of the signaling cascades trigger an increase of the intracellular calcium concentration ([Ca^2+^]_i_), which, in turn, activates signaling pathways and ion channels such as the Ca^2+^-activated K^+^ channel, Ca^2+^-activated chloride channel and transient receptor potential (TRP) channel [10,11]. Therefore, Ca^2+^ acts as a second messenger in numerous primary functions of VSMCs such as vascular tone, cell proliferation, vasculogenesis or vasoactive factors release, as reviewed in [12].

The endothelium is the primary tissue responsible for the regulation of VSMCs’ contractility, vascular wall permeability, angiogenesis, triggering coagulation and fibrinolysis and the regulation of the vascular tone and vessels diameter, as reviewed in [13,14]. In all these mechanism, endothelium production of nitric oxide (NO), prostaglandin and the secretion of vasoactive agonists play a critical role [15]. In the case of VSMCs, they are normally quiescent and contractile, although they change to a proliferative and migratory state in certain conditions, such as arterial injury or inflammation [16]. This change is considered a characteristic step in the pathogenesis of multiple vascular diseases and is associated with ion channels’ plasticity and a rise in [Ca^2+^]_i_, which activate Ca^2+^-dependent factors of transcription [17]. Among other channels, TRP channels seem to be implicated in the [Ca^2+^]_i_ enhancement during VSMCs’ proliferation and vascular remodeling [18].

This review is organized into two main sections. The first provides a brief overview of the TRP channel expression and physiological function in vascular beds. The second considers the role of TRPC and TRPV channels in vascular remodeling and disorder, such as systemic and pulmonary arterial hypertension, atherosclerosis and restenosis, as indicated in Figure 1. In this review, we will focus in particular on the roles of TRPC and TRPV, which are among the most important TRPs in vascular beds.

## 2. TRPs in Vascular Beds and Their Role in Vascular Function

TRP channels are a superfamily of Ca^2+^-permeable cation channels that exhibit a common structure: six putative transmembrane domains (TM1–TM6) and one loop located between TM5 and TM6 shaping the cation-permeable pore region. According to the function and genetic similitude, TRP channels are classified into the canonical channel (TRPC), vanilloid-related channel (TRPV), melastatin-related channel (TRPM), ankyrin-related channel (TRPA), mucolipin-related channel (TRPML) and polycystin-related channel (TRPP) (for a review, see [19,20]). TRP channels are permeable to monovalent and divalent cations, mainly to Na^+^ and Ca^2+^, showing a high permeability for Ca^2+^. TRP channels contribute to the modulation of [Ca^2+^]_i_, either directly by supporting the Ca^2+^ influx through the plasma membrane or indirectly by modulating the resting membrane potential that controls Ca^2+^ entry through voltage-dependent Ca^2+^ channels (VDCCs) [21]. One of the most characteristic features of TRP channels is that they can be modulated by numerous stimuli, such as temperature fluctuations, mechanical stress, neurohormonal signals, H^+^, Ca^2+^ and Mg^2+^ ions and intracellular ligands, as well as vasoactive factors like angiotensin-II, ATP, thrombin and endothelin-1 (ET-1) [22,23,24].

Both ECs and VSMCs express different TRP channels. In the case of ECs, they have been described as expressing TRPC1/3-7, TRPV1/2/4, TRPM1-4/6-8, TRPA1 and TRPP1 at the transcript or protein levels. However, not all TRP isoforms are expressed in ECs. The pattern is highly variable in relation to the tissue and the animal species, as reviewed in [13,25]. Furthermore, different studies demonstrate that VSMCs express mostly TRPC1/3-7, TRPV1-4, TRPM2-4/6/7/8, TRPA1 and TRPP1/2 [26,27,28]. Generally, the level and the expression of the TRP subfamily depend on cells and microenvironment requirements. Given the number of roles performed by TRP channels in the vascular function, we will here review, in particular, the role of TRPC and TRPV in the myogenic response and vascular tone.

### 2.1. Myogenic Response

The myogenic response is conducted by VSMCs and is one of the main pressure-sensitive mechanisms that keeps a regular blood flow independent of changes in the arterial blood pressure [29]. There have been studies on whether TRPs have an essential role in this vascular function.

TRPCs: The role of TRPC1 in the myogenic function has been extensively investigated ever since Maroto et al. [30] suggested its role as a mechanosensitive channel. However, its implication in myogenic regulation remains unclear. Dietrich et al. [31] demonstrated that the pressure-induced constriction of cerebral arteries, as well as the hypo-osmotic swelling and positive pipette pressure-induced cation currents of VSMCs, were not impaired in a TRPC1-deficient mouse model. The TRPC1-knock out (KO) mouse did not present significant differences in the myogenic response as compared to a wild-type mouse, suggesting that TRPC1 is not necessary for myogenic tone regulation. Later, Anfinogenova et al. [32] demonstrated that agonists of the Gq/11-coupled receptor can modulate TRPC1-like currents in cerebral arterial smooth muscle cells but that they do not enhance their mechanosensitivity or the ability of intact cerebral arteries to respond to intravascular pressure changes, concluding that TRPC1 is not essential for myogenic tone regulation. Furthermore, Reading et al. [33] showed an unaffected response to increased intravascular pressure in cerebral arterial VSMCs when they suppressed TRPC3 using antisense oligodeoxynucleotides. Thus, TRPC3 would not be involved in the myogenic response either [33]. In contrast, independent studies demonstrated that TRPC6 may be required for myogenic tone regulation. Dietrich et al. [34] used TRPC6-KO mice to demonstrate that agonist-induced aortic and mesenteric contractility was increased and that these KO mice exhibited a higher mean arterial blood pressure, indicating a role of TRPC6 in vascular tone regulation. This finding was supported by Schnitzler et al. [35], who proposed that TRPC6 is important to the myogenic response, not as a mechanosensitive channel but as a channel activated by G-protein coupled receptors [35]. Moreover, the role of TRPC6 in the myogenic response was proposed in rat posterior cerebral arteries by blocking the channel using SKF96365 (although this drug is not specific for TRPC6 [36]) and the corresponding siRNA [37]. Gonzales et al. [38] further supported this evidence (suggesting that TRPC6 interacts with TRPM4) by using an in situ proximity ligation assay (PLA) to promote membrane depolarization induced by mechanical stimuli [38]. Hence, there is an agreement that TRPC6 participates in the myogenic response, but that TRPC1 and TRPC3 do not.

TRPVs: An early report demonstrated that the presence of TRPV1 in vascular sensory C-fibers served as a trigger of myogenic constriction in mesenteric arteries [39]. This study established that the elevation of intraluminal pressure was associated with the generation of the arachidonate metabolite, 20-HETE, which in turn activated TRPV1 on C-fiber nerve endings, resulting in depolarization and consequent vasoactive neuropeptide release. Therefore, TRPV1 likely contributes to myogenic constriction, not as mechanoreceptors in mesenteric arteries, but as a sensor in C-fibers that mediates Bayliss’ myogenic constriction of arteries [39]. In the case of TRPV2, McGahon et al. [40] demonstrated that tranilast included in the patch pipette to inhibit TRPV2 channels or specific pore-blocking with an TRPV2 antibody prevented a negative pressure-induced current increase, indicating that TRPV2 may be required for the myogenic response of rat retinal arterioles [40]. Interestingly, in 2017, a study by Soni et al. [41] proposed that TRPV4 was vital for neonatal pig preglomerular VSMCs’ myogenic autoregulation, since its inhibition with HC067047 reversed pressure-induced membrane depolarization, [Ca^2+^]_i_ elevation and constriction in renal preglomerular arteries. Therefore, TRPV1/V2 and V4 likely play a role in myogenic tone regulation.

### 2.2. Vascular Tone

VSMCs and ECs are responsible for regulating the vascular tone, which is essential to keep the blood flow circulating. The regulation of the vascular tone activity includes the effect of a variety of stimuli, such as hormones, neurotransmitters, endothelium-derived factors and the blood pressure itself [42]. Different subfamilies of TRP have shown significant involvement in agonist-induced vasoconstriction (Figure 1A) and the major role of TRP channels in smooth muscle contraction is often ascribed to the activation of VDCCs through cell depolarization, the main regulators of vascular tone [21].

TRPCs: There is evidence indicating that TRPC channels modulate the membrane potential and [Ca^2+^]_i_ and that they play a key role in the vascular tone, as reviewed in [43]. Several studies have suggested that TRPC1 activation induced VSMCs’ hyperpolarization and attenuated arterial vasoconstriction. For instance, Kwan et al. [44] demonstrated that TRPC1 activation promoted the hyperpolarization of rat aorta VSMCs. They determined that TRPC1 that was physically associated with BK_Ca_ (large conductance potassium channels sensitive to Ca^2+^) and that Ca^2+^ influx through TRPC1 activated BK_Ca_, which induced membrane hyperpolarization. Kochukov et al. [45] also showed that aorta from TRPC1-KO mice, but not TRPC3-KO ones, exhibited an increased phenylephrine-evoked vasoconstriction due to the reduced role of BK_Ca_ channels. However, Schmidt et al. [46] proposed that TRPC1 was a negative regulator of endothelial Ca^2+^-activated K^+^ channels (K_Ca_) in ECs and that it thereby contributed to blood pressure regulation. The authors observed an augmented endothelium-derived hyperpolarizing factor (EDHF)-mediated vasodilatation in the carotid arteries of TRPC1- but not of TRPC6-KO mice. In addition, the TRPC1-KO mice showed a reduced systolic blood pressure [46]. Furthermore, we have recently demonstrated that TRPC1 and Orai1, the pore-forming subunit of the store-operated Ca^2+^ channel (SOCC), interacted with the voltage-dependent Ca_v_1.2 L-type Ca^2+^ channels (LTCC) to control agonist-induced aorta and coronary artery contraction [47,48]. We provided evidence showing that serotonin (5-HT), ET-1 and thapsigargin (TG) enhanced the basal interaction between Ca_v_1.2, Orai1 and/or TRPC1, which highlighted the relevance of these channels to vascular tone regulation [47,48].

Similarly, the role of other TRPCs in vascular tone regulation was assessed using KO mice or antibodies. Actually, phenylephrine-induced vasoconstriction was reduced in TRPC3-KO mice when compared to wild type mice [49]. TRPC3 and C7 blockade, using antibodies, also inhibited cation channel currents induced by ET-1 in rabbit coronary artery myocytes [50]. TRPC4-KO mice had an attenuated SOCC current in ECs and therefore exhibited a reduced agonist-induced vasorelaxation [51]. Meanwhile, the silencing of TRPC6 decreased hypoxia-induced pulmonary vasoconstriction [52].

TRPVs: There are only a few reports that have studied the role of TRPV in vascular tone regulation and most of them have demonstrated their relevance in relation to ECs’ function. Actually, endothelial TRPV1 participated in NO release, which induced rat mesenteric and pig coronary arteries’ relaxation [53,54]. Zhang et al. [55] used TRPV4-KO mice to demonstrate the endothelial TRPV4 relevance in a vasodilation induced by acetylcholine. Likewise, Sonkusare et al. [56] proposed TRPV4 as part of the ion channel cluster formed by intermediate (IK)- and small (SK)-conductance Ca^2+^-sensitive K^+^ channels, whose activation caused the maximal vasodilatation in the vascular endothelium of resistance arteries.

Altogether, these studies suggest that the expression of TRPC1/3 and 7, or of TRPV1 and 4, either in VSMCs or ECs, might be required for vascular tone regulation.

## 3. TRPC and TRPV Channels in Vascular Diseases

Vascular remodeling relies on molecular, cellular and structural changes in the blood vessel, which happen in response to chronic alterations in the blood flow or in response to vessel wall injury. This process is brought about by some cardiovascular diseases, like systemic or pulmonary hypertension, atherosclerosis or restenosis [57]. Ca^2+^ signaling and TRP channels stand as significant regulators of the VSMCs transition from a normal physiological phenotype to a dysfunctional diseased phenotype (Figure 1, Table 1).

### 3.1. Systemic Hypertension

Hypertension is a complex disease caused by genetic and environmental interactions characterized by a persistent increase in the vascular tone and an augmented blood pressure. The main pathological change in hypertension is vascular remodeling, which is associated with an increase in the media thickness due to VSMCs hyperplasia and/or hypertrophy [58]. Other key pathological features of hypertension are an impaired bioavailability of NO by ECs and the secretion of vasoactive agonists, which in turn modulate Ca^2+^ signaling in VSMCs [59]. Different isoforms of TRP channels contribute to hypertension-evoked vascular remodeling and enhanced vasoconstriction responses thanks to the alteration of [Ca^2+^]_i_ and VSMCs proliferation [60,61].

TRPCs: Several studies used mouse model of essential hypertension or spontaneously hypertensive rats (SHRs) to demonstrate the significant upregulation of TRPC isoforms in aorta, mesenteric or carotid arteries [62,63,64]. In mesenteric arterioles from SHRs, the expression of TRPC1, C3 and C5 was augmented and correlated with an increase in norepinephrine-induced oscillations of the arterial vascular tone, known as vasomotion [64]. Antibodies against TRPC1, C3 or C5 inhibited a norepinephrine-induced Ca^2+^ increase and vasomotion in these resistance arteries, confirming the implication of these TRPCs in this process. Lin et al. [63] also showed that TRPC1, C3 and C6 were upregulated in the carotid artery of SHRs. Nevertheless, they demonstrated that only TRPC1 and C6 were involved in the increase of the medial thickness, lumen diameter, medial area, collagen deposition and medial VSMCs hyperplasia, typical indicators of arterial remodeling observed with age and systolic blood pressure [63]. Meanwhile, TRPC3 upregulation was found in eight-week SHRs without carotid arterial remodeling, but not in 18-week SHRs with arterial remodeling, indicating that TRPC3 may be important for hypertension development. In contrast, a previous report by Noorani et al. [65] found an increased TRPC3 and decreased TRPC1 expression at the protein levels in the carotid artery of SHRs. The authors proposed that TRPC3 overexpression in hypertension led to a greater Ca^2+^ and Na^+^ influx, depolarization and consequent activation of VDCCs, which enhanced carotid artery contractility [65]. Recently, Alvarez–Miguel et al. [62] effectively demonstrated that hypertension promoted changes in TRPC3 and C6 heteromultimeric assembly, which favored VSMCs depolarization. In accordance with these findings, an increased TRPC3 expression was detected in purified mitochondria in the vasculature of SHRs when compared to Wistar–Kyoto rats. Moreover, TRPC3-KO mice showed a reduced angiotensin II-induced reactive oxygen species (ROS) production, which suppressed vasoconstriction and decreased blood pressure [66]. TRPC3 upregulation was also observed in the mesenteric arteries and in aortic tissue of SHRs, where it was associated with exacerbated vasoconstrictions to ET-1 and angiotensin II during hypertension [67,68]. Interestingly, the upregulation of TRPC3 was also detected in monocytes from SHRs [69] and from patients with essential hypertension [61]. This overexpression correlated with an increase in the release of pro-inflammatory cytokines, such as IL-1β and TNF-α, in patients with essential hypertension when compared to normotensive control subjects [70]. In the case of TRPC6, an early study by Dietrich et al. [34] observed an elevated blood pressure and enhanced agonist-induced contractility in the aortic and cerebral arteries of TRPC6-KO mice. However, independent studies demonstrated that TRPC6 was upregulated under hypertension, using for example deoxycorticosterone acetate-salt hypertensive rats model [34] or the Milan hypertensive strain (MHS) [71]. For instance, Bae et al. [72] showed that the TRPC6 expression was enhanced in the mesenteric artery of DOCA-salt hypertensive rats, suggesting that aldosterone evoked TRPC6 upregulation, receptor-operated Ca^2+^ entry (ROCE) and consequently hypertension development.

TRPVs: Studies using TRPV1, V4-KO mice and pharmacological compounds that selectively target both channels, suggested that they regulate blood pressure and play a protective role against hypertension [108,109]. In Dahl salt-sensitive rats, TRPV1 inhibition, using capsazepine, enhanced the blood pressure remarkably; meanwhile, its activation with capsaicin decreased the mean arterial pressure in a dose-dependent manner [74]. Zhang et al. [73] also showed that the expression of TRPV1 in mesenteric arteries and the kidney was downregulated in this rat model of hypertension. Moreover, TRPV1 activation by capsaicin inhibited hypertension-induced VSMCs phenotypic switching, reduced the intracranial arteriole remodeling [73] and improved the endothelial production of NO in rats, which prevented hypertension [75,76]. Marshall et al. [110] further demonstrated that the blood pressure of wild-type mice fed with a high-fat diet was higher than TRPV1-KO mice’s blood pressure. Regarding the TRPV4’s role in hypertension, it was mainly related to its vasodilatory actions through the endothelium [77]. However, it has also been reported that VSMCs hyperpolarization contributed significantly in mediating TRPV4-dependent vasodilation [78]. The administration of L-NAME to inhibit nitric oxide synthase (NOS) promoted an increase of blood pressure in TRPV4-KO mice, indicating that TRPV4 mediated a vasodilatory compensatory mechanism to regulate blood pressure [79]. Other studies have demonstrated that the activation of TRPV4 with 4a-PDD reduced the basal and elevated the blood pressure in normal and high salt intake rats, suggesting that TRPV4 activation and upregulation may constitute a counter-regulatory mechanism to prevent salt-induced increases in blood pressure in Dhal-resistant rats [80]. Interestingly, Diaz–Otero et al. [81] demonstrated that the activation of the mineralocorticoid receptor impaired TRPV4-mediated relaxation in parenchymal arterioles and reduced cognitive function during hypertension. This study showed that mRNA of TRPV4 was reduced in cerebral arteries in angiotensin II-induced hypertensive mice and was recovered by mineralocorticoid receptor activation.

Therefore, there is a general consensus regarding the deleterious role of different isoforms of TRPC in systemic hypertension and their implication in arterial remodeling and in greater responses to vasoactive agonists. By contrast, TRPV1 and V4 seem to play a protective role against hypertension (Table 1).

### 3.2. Pulmonary Arterial Hypertension

Pulmonary arterial hypertension (PAH) is a hemodynamic disorder caused by pulmonary vasculature remodeling, vasoconstriction and thrombosis, responsible for a progressive elevation of the pulmonary arterial pressure that leads to right heart failure, and even death [93,111]. Chronic hypoxia is the main cause of pulmonary artery hypertension and is contributed by pulmonary media hypertrophy caused by the proliferation of pulmonary artery smooth muscle cells (PASMCs), which narrows the intraluminal diameter and increases resistance to blood flow [28,112].

TRPCs: TRPC channels are considered an alternative and/or additional Ca^2+^ influx pathway in PASMCs and pulmonary ECs (Figure 1B). As in other VSMCs, TRPC1 and STIM1 functionally associate in order to mediate store-operated Ca^2+^ entry (SOCE) in PASMCs [113]. Previous studies showed that TRPC1, C3 and C6 were expressed at the mRNA and protein levels in primary human PASMCs. Moreover, the chronic exposure to hypoxia of intralobar pulmonary arteries increased the expression of TRPC1 and C6, accompanied by enhanced SOCE and ROCE [113,114]. Similarly, bone morphogenetic protein 4 induced a SOCE increase, proliferation and migration of human PASMCs through TRPC1, C4 and C6 upregulation [82,83]. Interestingly, Alzoubi et al. [115] demonstrated that TRPC4 inactivation in rats decreased the acetylcholine-induced Ca^2+^ increase in PASMCs and reduced the severity of the occlusive pulmonary arteriopathy in a rat model of PAH exposed to Sugen 5416/hypoxia/normoxia (Su/Hx/Nx). This fact shows that the loss of TRPC4 may prevent pulmonary vascular remodeling. Moreover, Yu et al. [116] determined that TRPC6 was highly expressed in the PASMCs of patients with idiopathic pulmonary arterial hypertension (iPAH), where it was involved in PASMCs proliferation. In accordance with this data, a unique genetic variant in the promoter of the gene *TRPC6* was discovered in iPAH patients [86]. Therefore, TRPC6 was suggested as a therapeutic target for PAH. Actually, Bosentan, an endothelin receptor inhibitor used clinically for iPAH patients, decreased PASMCs’ growth and proliferation through TRPC6 downregulation [88]. Furthermore, Sildenafil, another drug used widely to treat PAH, efficiently decreased TRPC1 and TRPC6 expression in the distal pulmonary arteries of chronically hypoxic rats and inhibited the chronic hypoxia-induced increase in basal [Ca^2+^]_i_ and SOCE in PASMCs [84,85].

TRPVs: Independent studies suggested that TRPV1, V3 and V4 also play a role in PASMCs’ proliferation and PAH. TRPV1 activation increased [Ca^2+^]_i_ in PASMCs, which enhanced vascular contraction, proliferation and migration through NFAT and CREB activation [92]. The incubation of PASMCs with hypoxia activated TRPV1 and V4, which increased PASMCs’ migration and cytoskeleton reorganization [117]. Similarly, PASMCs isolated from iPAH patients presented an overexpression of TRPV1 and V4 when compared with healthy subjects [89,90]. Furthermore, TRPV4 knockdown with siRNA significantly attenuated the shear stress-induced increase of [Ca^2+^]_i_ in iPAH-PASMCs [89], confirming the role of TRPV4 as a mechanosensitive channel sensible to flow shear stress. Moreover, an increased Ca^2+^ entry through TRPV4 contributed to an enhanced PASMCs contraction, migration and proliferation triggered by chronic hypoxia [118]. More recently, TRPV4 was studied in pulmonary arterial adventitial fibroblasts using TRPV4-KO mice, siRNA and the pharmacologic inhibition of TRPV4 [93]. This study revealed that TRPV4 played an important role in the proliferation, migration and synthesis of the extracellular matrix, hallmarks of the pathogenesis of PAH [93]. On the other hand, TRPV3’s role in PAH was examined by Zhang et al. [91], who showed that TRPV3 was upregulated in the pulmonary vessels of PAH humans when compared to control subjects. They further demonstrated that TRPV3 participated in the hypoxia-induced proliferation of PASMCs and pulmonary vascular remodeling through the PI3K/AKT pathway [91].

In summary, PAH is associated with the upregulation of TRPC1/C3/C4/C6 and TRPV1/V3/V4 (Table 1), which play an important role in the increase of Ca^2+^ influx in PASMCs and in their proliferation and migration, which are critical steps for pulmonary artery remodeling.

### 3.3. Atherosclerosis

Atherosclerosis is a chronic inflammatory disease characterized by ECs and VSMCs proliferation and migration. During this pathogenesis, inflammatory cells, mainly monocytes and neutrophils, are recruited to the vascular wall (the site of inflammation), suffering a pathological growth that leads to atheroma plaque formation [119]. Subsequently, in the atheroma plaque, macrophages produce inflammatory cytokines and engulf lipids in an attempt to minimize the pathological accumulation of cholesterol and other lipids. In the past two decades, TRP channels have been implicated in both aspects of the disease: the proliferation of cells in the vascular wall and the pathophysiology of macrophages (Figure 1C and Table 1) [120,121].

TRPCs: Some studies reported that the reactivity of ECs to ET-1 was increased with hypercholesterolemia, considered a potent trigger in atherogenesis [122]. Bergdahl et al. [123] and Ingueneau et al. [124] demonstrated that cholesterol influenced the vascular reactivity to ET-1, affecting the caveolar localization of TRPC1. Similarly, the specific deletion of TRPC3 from bone marrow cells reduced macrophages’ presence in the atheroma plaque [125] and the specific suppression of this channel in macrophages reduced necrosis inside the atheroma plaque [96]. In contrast, the overexpression of TRPC3 in the endothelium exacerbated endothelial inflammation and macrophage infiltration, resulting in an increased burden of advanced aortic atherosclerosis [94,95]. Interestingly, a recent study by Min et al. [97] used a high-fat diet-fed apolipoprotein E knock-out (ApoE^-^-KO) mice model of atherosclerosis to demonstrate that microRNA-26a overexpression, through TRPC3 inhibition, suppressed inflammatory responses and the NF-κB pathway, promoting cell viability and inhibiting apoptosis in oxidized low-density lipoprotein (ox-LDL)-stimulated ECs. A previous study by Zhang et al. [98] also demonstrated that this microRNA-26a was downregulated in the aortic intima of ApoE-KO mice, where it specifically inhibited TRPC6 expression and ECs apoptosis, indicating that TRPC3 and C6 suppression may prevent atherosclerosis progression.

TRPVs: Compelling evidence points to the fact that TRPV1 and V4 play a role in atherosclerosis. The long-term activation of TRPV1 by capsaicin significantly reduced lipid storage and atherosclerotic lesions in the aortic sinus and in the thoracoabdominal aorta of ApoE-KO mice, but not of ApoE and TRPV1 double KO mice [99]. TRPV1 activation also inhibited VSMCs proliferation in an atherosclerosis context [100] and prevented inflammation and oxidation [87,101]. Moreover, the protein level of TRPV1 was markedly higher in the aorta of ApoE-KO than in wild-type mice and was upregulated in ox-LDL-treated bone-marrow-derived macrophages. Therefore, the role of TRPV1 was linked to the lipid metabolism and inflammatory responses of macrophage-foam cells [126]. Similarly, Goswami et al. [102] demonstrated that TRPV4 genetic ablation or its pharmacologic inhibition blocked ox-LDL-induced macrophage foam cell formation and prevented pathophysiological matrix stiffness, indicating that TRPV4 activation might prevent atherosclerosis progression.

Altogether, these studies indicate that TRPC3/C6 and TRPV1/V4 in particular are involved in different aspects of atherosclerosis progression, while more data are needed to confirm the role of other TRPC and TRPV isoforms.

### 3.4. Restenosis

Restenosis is the recurrence of stenosis in vascular blood vessels that occurs mainly after angioplasty [127]. Vascular injury is characterized by an endothelial denudation that leads an inflammatory response; this results in the onset of several proliferative processes, including VMSCs switching from contractile to synthetic phenotypes, which contributes to the development and progression of restenosis [128,129]. This process is regulated by changes in Ca^2+^ signaling, ion channels and transcription factors activation [17,130,131,132]. However, only few reports have addressed the role of TRP channels in restenosis and most of them have focused on TRPC; meanwhile, there is no data regarding TRPV implication in restenosis (Figure 1D and Table 1). Bergdahl et al. [103] showed that the balloon-induced dilatation of human internal mammary arteries enhanced the plasticity of TRPC expression. Specifically, the authors observed a significant increase in the expression of TRPC1 and TRPC6 at the mRNA level, which correlated with cellular Ca^2+^ handling. This finding was supported by an experiment done in pig coronary arteries [104] and in human vein samples, in which the upregulation of TRPC1 was also observed after angioplasty [106]. Interestingly, TRPC1 antibodies prevented neointima progression in human vein samples cultured in vitro [106]. Recently, Jia et al. effectively showed that TRPC1 interacted with Orai1 and Homer to promote neointima formation and stenosis in balloon-injured rat carotid arteries [105]. Moreover, Koenig et al. [107] demonstrated, using a novel ex vivo organ culture model based on stent implantation in aortic constructs, that Pyr3 inhibited the stent-induced upregulation of TRPC3 expression and the proliferation of human coronary smooth muscle cells, a particular parameter of stent-induced vascular remodeling. Therefore, TRPC1,C3, and C6 seem to play a role in the restenosis of different arteries; meanwhile, information regarding the role of TRPV is lacking.

## 4. Conclusions

Vascular remodeling and consequent disorders are highly prevalent diseases that cause significant morbidity and mortality. Increasing number of studies have shed light on the contribution of TRPC and TRPV to ECs’ and VSMCs’ physiological and pathological functions. Studies with animal models of common vascular diseases, like systemic and pulmonary hypertension, atherosclerosis and restenosis, which use KO mice of different TRP isoforms, suggest the translational potential role of TRPC and TRPV channels. Previous cutting-edge studies have suggested that these channels might be major regulators of cellular remodeling through their ability to modulate transcriptional programs that drive cell proliferation and migration, and even cytokine secretion [18,70,92]. As reviewed above, there is general agreement that most isoforms of TRPCs are implicated in cellular remodeling, a critical step for vascular disease progression, which occurs in systemic and pulmonary hypertension, atherosclerosis and restenosis. Meanwhile, TRPV1, V3 or V4 likely play a protective role in systemic hypertension and atherosclerosis, since they participate in NO release via endothelium and metabolism lipid regulation [87,101,108,109]. In contrast, TRPV1 and V4 activation seem critical for hypoxia-induced Ca^2+^ increase and further PASMCs proliferation and migration [92,117]. Nevertheless, future studies are urgently needed to address the downstream targets and mechanisms modulated by these channels, as well as the upstream regulators leading to changes in their expression and function during disease.

## Figures and Tables

**Figure 1 ijms-21-06125-f001:**
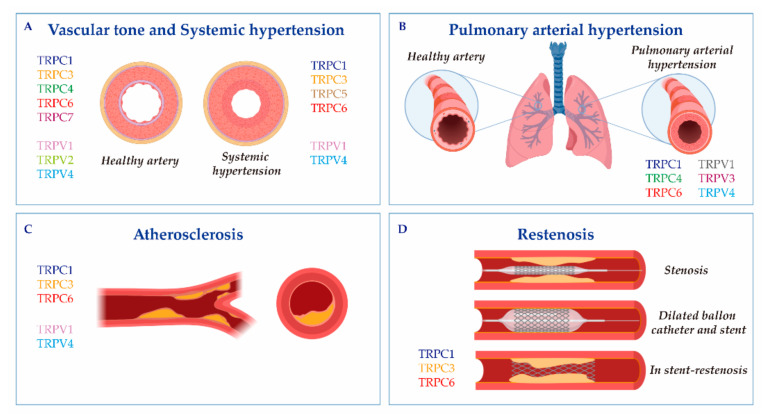
The cartoon indicates Transient receptor potential canonical (TRPC) and vanilloid (TRPV) isoforms involved in (**A**) vascular tone and systemic hypertension; (**B**) pulmonary arterial hypertension; (**C**) atherosclerosis; and (**D**) restenosis. Vascular responses and disorders depend either on TRP activation in the endothelium and vascular smooth muscle cells, or on critical changes in their expression.

**Table 1 ijms-21-06125-t001:** List of TRPC and TRPV isoforms involved in vascular disease. ApoE^—^KO, Apolipoprotein E knock-out mice; ECs, Endothelial cells; NF-κB, Nuclear factor kappa B; PAH, Pulmonary arterial hypertension; PASMCs, Pulmonary artery smooth muscle cells; ROS, Reactive oxygen species; SHRs, Spontaneously hypertensive rats; TRPC, Transient receptor potential canonical channel; TRPV, Transient receptor potential vanilloid-related channel; VSMCs, Vascular smooth muscle cells.

Disease		Trp Channel and Mechanism
**Systemic hypertension**	TRPC TRPV	Upregulation of TRPC1, C3, C5 and C6 in arteries SHRs [62,63,64]. They are involved in agonist-induced Ca^2+^ increase and vasomotion in arteries from SHRs [64]. TRPC1 and C6 are implicated in arterial remodeling in SHRs [63]. TRPC3 plays a role in hypertension development [63], ROS production and blood pressure [66]. TRPC6 is involved in aldosterone-induced Ca^2+^ influx and hypertension [72]. TRPV1 is downregulated in Dahl salt-sensitive rats [73]. TRPV1 activation decreases blood pressure [73,74], arterial remodeling [73], and improves nitric oxide production [75,76]. TRPV4 is implicated in VSMCs hyperpolarization and blood pressure regulation [77,78,79,80,81].
**Pulmonary arterial hypertension**	TRPC TRPV	Upregulation of TRPC1, C3, C5 and C6 in the human PASMCs of patients with idiopathic PAH [81,82,83,84,85,86]. Upregulation of TRPC1, C4 and C6 increases Ca^2+^ influx, proliferation and migration of PASMCs [82,83]. TRPC4 inactivation prevents pulmonary vascular remodeling [87]. TRPC6 overexpression promotes PASMCs’ proliferation [86,88]. Upregulation of TRPV1, V3 and V4 in the PASMCs of patients with idiopathic PAH [89,90,91]. TRPV1 and TRPV4 activation evokes Ca^2+^ influx, proliferation and contraction of PASMCs [85,89,90,92]. TRPV4 plays a role in fibroblast proliferation and the synthesis of the extracellular matrix in lungs [93]. TRPV3 participates in hypoxia-induced PASMCs’ proliferation and remodeling [91].
**Atherosclerosis**	TRPC TRPV	TRPC3 overexpression increases ECs’ inflammation and macrophage infiltration [94,95]. TRPC3 deletion in macrophages reduces their presence in the atheroma plaque [96]. TRPC3 inhibition suppresses the NF-κB pathway, promotes cell viability and inhibits apoptosis in ECs [97]. TRPC6 inhibition prevents ECs apoptosis [98]. TRPV1 activation reduces atherosclerotic lesions in the aorta of ApoE-KO mice [99], inhibits VSMCs proliferation [100] and prevents inflammation and oxidation [87,101]. TRPV4 activation prevents atherosclerosis progression [102].
**Restenosis**	TRPC	Overexpression of TRPC1 and TRPC6 correlates with Ca^2+^ handling in balloon-injured human internal mammary arteries [103], pig coronary arteries [104] and rat carotid arteries [105]. TRPC1 antibody prevents neointima progression in human veins [106]. Stent implantation in the aorta promotes TRPC3 upregulation [107].

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
