# Peer review of "TRPC and TRPV Channels’ Role in Vascular Remodeling and Disease"

_ijms, 2020, doi:10.3390/ijms21176125_

Round 1

Reviewer 1 Report

This manuscript covers an important topic - TRPC and TRPV channels role in vascular remodeling and disease by collecting information from several studies about the implication of TRPC and TRPV in physiological and pathological processes of some frequent vascular diseases. The strengths of the paper include the public health importance of the topic and the comprehensive data source. However, the paper could be substantially strengthened by addressing the following small concerns. 

-This review is very well written by including information from different studies from physiological aspect toward vascular disorders, however adding a table which summarize the studies will make it easier to follow,

-Authors just mentioned several studies in each section without clear conclusion. Adding some sentences as conclusions at the end of each section will make the review more comprehensive,

Author Response

We wish to thank our valuable reviewer for his constructive comments. According to his recommendation we performed changes that certainly improve substantially our manuscript.

  • We added a table which summarize the finding of the studies discussed in this revision.
  • We also added short conclusion to highlight the main finding in each section.

Reviewer 2 Report

The manuscript “TRPC and TRPC channels role in vascular remodeling and disease” by Marta Martin-Bornez at al summarize the current knowledge and understanding of the roles that played by TRPC and TRPV ion channels in physiological and pathological processes of some frequent vascular diseases. Authors first briefly introduce the current research results about TRP ion channel expression and their physiological functions in vascular beds, such as myogenic response and vascular tone. The authors then spent much of their efforts focusing on the various effects and implications of TRPV and TRPC ion channels in some frequent vascular diseases – systemic hypertension, pulmonary arterial hypertension, atherosclerosis and restenosis. The manuscript is well written, message is clear, and easy to understand for general audience. In the end authors also point out the key issues needs to be addressed in the future.

I only have two minor comments/suggestions for authors:

  1. Figure 1, panel B: it would be easier for general audience to understand if the healthy pulmonary artery and pulmonary arterial hypertension can be labeled, just like that in panel A.
  2. P5, line170, “Vascular remodeling relies to ……” change to “Vascular remodeling relies on ……”

Author Response

Thank you very much for your kind comments. According to your recommendation we performed changes that are highlighted in this revised version of this manuscript.